# Multivariate method for prediction of fumonisins B$_1$ and B$_2$ and zearalenone in Brazilian maize using Near Infrared Spectroscopy (NIR)

Denize Tyska[1]☯, Adriano Olnei Mallmann[2]☯, Juliano Kobs Vidal[1]☯, Carlos Alberto Araújo de Almeida[1]☯, Luciane Tourem Gressler[3]☯, Carlos Augusto Mallmann[1]*

1 Department of Animal Health and Reproduction, Laboratory of Mycotoxicological Analyses (LAMIC), Federal University of Santa Maria (UFSM), Santa Maria, Rio Grande do Sul, Brazil, 2 Pegasus Science, Santa Maria, Rio Grande do Sul, Brazil, 3 Independent Veterinary Researcher, Santa Maria, Rio Grande do Sul, Brazil

☯ These authors contributed equally to this work.
* mallmann@lamic.ufsm.br

**Data Availability Statement:** All relevant data are within the manuscript and its Supporting information files.

## Abstract

Fumonisins (FBs) and zearalenone (ZEN) are mycotoxins which occur naturally in grains and cereals, especially maize, causing negative effects on animals and humans. Along with the need for constant monitoring, there is a growing demand for rapid, non-destructive methods. Among these, Near Infrared Spectroscopy (NIR) has made great headway for being an easy-to-use technology. NIR was applied in the present research to quantify the contamination level of total FBs, *i.e.*, fumonisin B$_1$+fumonisin B$_2$ (FB$_1$+FB$_2$), and ZEN in Brazilian maize. From a total of six hundred and seventy-six samples, 236 were analyzed for FBs and 440 for ZEN. Three regression models were defined: one with 18 principal components (PCs) for FB$_1$, one with 10 PCs for FB$_2$, and one with 7 PCs for ZEN. Partial least square regression algorithm with full cross-validation was applied as internal validation. External validation was performed with 200 unknown samples (100 for FBs and 100 for ZEN). Correlation coefficient (R), determination coefficient (R$^2$), root mean square error of prediction (RMSEP), standard error of prediction (SEP) and residual prediction deviation (RPD) for FBs and ZEN were, respectively: 0.809 and 0.991; 0.899 and 0.984; 659 and 69.4; 682 and 69.8; and 3.33 and 2.71. No significant difference was observed between predicted values using NIR and reference values obtained by Liquid Chromatography Coupled to Tandem Mass Spectrometry (LC-MS/MS), thus indicating the suitability of NIR to rapidly analyze a large numbers of maize samples for FBs and ZEN contamination. The external validation confirmed a fair potential of the model in predicting FB$_1$+FB$_2$ and ZEN concentration. This is the first study providing scientific knowledge on the determination of FBs and ZEN in Brazilian maize samples using NIR, which is confirmed as a reliable alternative methodology for the analysis of such toxins.

**Funding:** This study was funded by the National Council for Scientific and Technological Development (CNPq), in the form of an fellowship of research productivity (PQ; grant number 310190/2018-5) awarded to CAM. Pegasus Science also provided support in the form of a pro-labore to AOM. The specific roles of this author are articulated in the 'author contributions' section. The funders had no role in study design, data collection and analysis, decision to publish, or preparation of the manuscript.

**Competing interests:** The authors have read the journal's policy, and the authors of this manuscript have the following competing interests to declare: AOM is a paid employee of Pegasus Science. This does not alter our adherence to PLOS ONE policies on sharing data and materials. There are no patents, products in development or marketed products associated with this research to declare.

## Introduction

Mycotoxins are toxic secondary metabolites produced by some filamentous fungi that grow naturally in many commodities around the world [1,2]. In Brazil, contamination of substrates by mycotoxigenic fungi is rather common, since climatic conditions favor their development and the production of mycotoxins. These substances are chiefly produced by three fungal genera: *Aspergillus*, *Penicillium* and *Fusarium* [3]. Of these, the genus *Fusarium* is of great importance for it encompasses the main producers of fumonisins (FBs), *F. verticillioides* and *F. proliferatum* [4,5] and zearalenone (ZEN), *F. culmorum*, *F. graminearum* and *F. crookwellense* [6–9]. These mycotoxins can have harmful effects on human and animal health [10–12].

FBs are natural contaminants of numerous cereals, especially maize, and occur globally at concentrations that generally induce subclinical poisoning in several species [13–17]. There are more than two dozen known FBs [18]; however, fumonisin $B_1$ ($FB_1$), fumonisin $B_2$ ($FB_2$) and fumonisin $B_3$ ($FB_3$) stand out for their toxic effects on humans and animals [14]. $FB_1$ is the most toxic and abundant of them all [19], representing about 70% of the total concentration in naturally contaminated food and raw materials, followed by $FB_2$ and $FB_3$ [20].

The detrimental impact of FBs in animals has been well described, and horses and pigs are the most susceptible species. In horses, FBs cause hemorrhagic-liquefactive brain lesions (equine leukoencephalomalacia) [21], while pigs are affected with pulmonary edema [22]. In humans, exposure to these toxins has been investigated in the context of neural tube defects and growth deficiency in children [12,23]. $FB_1$ is described as a potent carcinogen in laboratory and production animals [24], and several epidemiological studies have associated it with the development of esophageal cancer in humans [4,25,26]. As a result, the *International Agency for Research on Cancer* (IARC) has considered $FB_1$ as "possibly carcinogenic to humans" (group 2B) [27].

ZEN is a non-steroidal fungal estrogenic metabolite [28]. It occurs naturally in cereals such as wheat, barley, rice, oats and particularly in maize with a worldwide distribution [28–30]. Pigs are the domestic species with the greatest susceptibility to this toxin [31]. When ingested via diet, ZEN triggers several reproductive disorders in such animals [32], with the main clinical sign being known as hyperestrogenism syndrome [33,34]. In humans, however, studies assessing the possible effects of ZEN are scarce, but the occurrence of precocious puberty has been reported in children [35,36].

Maize has a high nutritional value and productive potential [37,38]. Brazil is one of its leading growers, third only to the United States and China [39]. This ingredient is very versatile in use; its largest share is destined to animal nutrition, especially poultry and swine, but it is also widely employed in the preparation of culinary dishes, being an important energy source for lower income populations [40].

Maize crops are greatly affected by fungi, especially the *Fusarium* species. Numerous studies have reported 90–100% prevalence of FBs in Brazilian raw maize [19–23]. For ZEN, positivity varies from one region to another, depending on climatic and grain storage conditions [41–43]. The National Health Surveillance Agency (ANVISA) established the legislation regarding the maximum tolerated limits (MTL) for mycotoxins in foods through Resolution No. 7 of February 18, 2011; the MTL for FBs and ZEN in maize grain for further processing is 5,000 and 400 µg.kg⁻¹, respectively [44]. In addition to individual mycotoxin contamination, many studies have demonstrated the possible synergistic effects of the co-occurrence of FBs and ZEN [45–47].

Monitoring the presence of these mycotoxins is crucial in view of their relevance. The classical methods of mycotoxins determination involve solvent toxin extraction processes and detection by chromatographic methods [48]. These techniques are very accurate, but

time-consuming and costly, thus hampering analyses of large samples and real-time decision making [49,50]. In search of fast techniques for the quantification of constituents in food samples, optical methods such as Near Infrared Spectroscopy (NIR) have made great headway [51].

This methodology is based on indirect measurements, since the generated spectral data are quite complex. The chemical composition of foods used in the manufacturing industry are altered when fungal infection and consequent contamination by mycotoxins occur. NIR has the potential to analyze these differences in specific ranges and build predictive models either through qualitative or quantitative methods [52–54]. Thus, it is necessary to use techniques that require calibrations with mathematical models and multivariate statistical tools in order to extract the analytical information from the corresponding spectra [55].

There is also a focus on the application of NIR as a classification method to discriminate fungal species, that is, to differentiate between toxigenic and non-toxigenic isolates [56,57]. NIR was evaluated as an indirect method that uses fungal counts as indirect markers to assess the risk of FBs contamination in maize samples; the obtained data was correlated with the limits established by the European legislation (4,000 μg.kg$^{-1}$) [58]. Moreover, the study analyzed the content of ergosterol and FB$_1$. The percentage of samples well classified in calibration and validation were 96 and 84%, respectively. Thus, the authors demonstrated the potential of NIR as a rapid method for screening maize samples according to their risk of FBs contamination [58].

One of the first suggested methodologies using NIR for fungal or mycotoxin determination used the quantification of ergosterol as a measure of living fungal biomass [59], and often associated it with the content of mycotoxins and fungal units. NIR is also applied to identify mycotoxins using chemometric methods of predictive quantification or classification in many raw materials: deoxynivalenol in single wheat kernels [60]; aflatoxins (AFs) in single whole maize kernels [61]; FBs in single maize kernels [62]; FB$_1$ in maize [63]; aflatoxin B$_1$ (AFB$_1$), ochratoxin A and total AFs in spices as red paprika [64]; AFB$_1$ in maize and barley [65]; and AFB$_1$ in red chili powder [66]. However, there is a paucity of information regarding the quantitative determination of ZEN in naturally contaminated maize.

Brazil has vast production as well as consumption of maize by both humans and animals, a cereal that is often contaminated by FBs and ZEN. In spite of that, no investigation has dealt with the analysis of such mycotoxins in Brazilian maize through NIR thus far. So, this study aims to fill such lack of scientific data by using NIR to quantitatively predict the concentration of total FBs, FB$_1$+FB$_2$, and ZEN in naturally contaminated Brazilian maize samples, and to assess the prediction potential in unknown samples.

## Materials and methods

### Maize samples

Six hundred and seventy-six maize samples were received, selected and analyzed at the Laboratory of Mycotoxicological Analyses (LAMIC) between 2018 and 2019. The samples were sent from different states of Brazil, thus originating from diverse climates and soils and making the data as representative of the whole country as possible. As the material assessed herein was part of LAMIC's routine analysis, no specific permission was required; furthermore, it was treated anonymously. The samples proceeded to milling, weighing, extraction and analyses. Grinding of each sample was standardized and performed in a Retsch ZM200 ultra-centrifugal mill with a particle size of approximately 1 mm in diameter. Then, a fraction was sent to the toxin extraction process and later to chromatographic analyses of mycotoxins by Liquid Chromatography Coupled to Tandem Mass Spectrometry (LC-MS/MS). Another fraction was used

for optical data collection, with the purpose of building the spectra library. This spectral set represented all the concentration variation normally found at the field level. Of the total number of samples, 236 were analyzed for FBs and 440 for ZEN. Two hundred unknown samples were used for external validation (one hundred for FBs and one hundred for ZEN); these elements were not included in the calibration database. The samples of the validation set were selected to represent the concentration range of the calibration set.

## Total fumonisins (FB$_1$+FB$_2$) measurements

A 3-g sample was added to 15 ml of acetonitrile:water solution (1:1, v/v) and vortexed for 20 min in a MA563 instrument (Marconi, Piracicaba, Brazil). The extract was then diluted in acetonitrile:water:formic acid solution (5:4:1, v/v/v), and 10 μl were injected into a 1200 Series Infinity HPLC (Agilent) coupled to an API mass spectrometer 5000 (Applied Biosystems) equipped with an ESI source in positive mode. Chromatographic separation was done at 40 ˚C using an Eclipse XDB-C8 column (4.6´150 mm, 5 μm particle diameter) (Agilent). The mobile phase gradient was composed of solutions of water:formic acid (95:5, v/v) (solution A) and acetonitrile:formic acid (95:5, v/v) (solution B) [67]. The limit of determination (LOD) and the limit of quantification (LOQ) for the assessed toxins were (in μg.kg$^{-1}$), respectively: FB$_1$, 10 and 125; and FB$_2$, 20 and 125.

## Zearalenone (ZEN) measurements

A method proposed by Berthiller et al. [68] was adapted to carry out ZEN analyses. A sample containing 3 g was added to 24 ml of a methanol:water solution (7:3, v/v) and vortexed for 20 min using a MA563 instrument (Marconi). The extract was then diluted in a methanol:water: ammonium acetate 1 M solution (90:9:1, v/v/v), and 10 μl was injected into a 1200 Series Infinity HPLC (Agilent) coupled to a 4000 QTRAP mass spectrometer (Applied Biosystems), equipped with an ESI source in positive mode. Chromatographic separation was performed at 40 ˚C with a Zorbax SB-C18 column (4.6´150 mm, 5 μm particle diameter) (Agilent). The mobile phase gradient consisted of solutions of methanol:water:ammonium acetate (90:9:1, v/v/v) (solution A) and water:ammonium acetate (90:10, v/v) (solution B).

## Near infrared spectroscopy

Spectra acquisition was performed by a Foss NIRS™ DS2500 equipment with silicon (400–1100 nm), lead sulfide (1100–2500 nm) detector and wavelength range from 400 to 2500 nm, 0.5 nm spectral resolution and 32 spectrum scans. The measurement mode data were acquired in reflectance and then converted to absorbance (- log$^R$) at the time of modeling. The *large sample cups* was the type of cell used for reading solid samples. Reading time of each spectrum was approximately 1 min. The spectrophotometer was connected to a computer that stored the spectra data collected using the ISISCAN nova program. The spectra file was converted into a JCAMP file, which was used for multivariate data analyses. The final spectral data were exported in order to be evaluated with FB$_1$, FB$_2$ and ZEN data and to perform chemometric analyses with Unscrambler v.9.7 software (CAMO, Norway).

## Statistical analyses

The calibration set was arranged in a matrix with 236 (FBs) and 440 (ZEN) rows and 4200 (variable independent) + 1 columns (variable dependent), combining spectral and chemical data for each sample. The absorbance value for each wavenumber was reported in the first 4200 columns, while the analytical concentration (μg.kg$^{-1}$) of FB$_1$, FB$_2$ and ZEN was reported

in the last column. Because spectral data is quite complex, this technique requires the use of several chemometric algorithms [69]. These tools aim at finding quantitative relationships between two sets of measured data [49].

The spectral information used in the model covered the full spectral range as it was not clear at that point in time in which specific wavelengths FBs would be present [62,63,70]. Several pre-treatments were tested in the spectral data, with the best accuracy in the model being chosen. Partial least squares (PLS) was used [71], considering that absorbance values of contiguous wavelength were collinear variables. This method is based on correlating two data matrices, one containing the new measurements, X (independent variables), and another with the values of the property of interest measured by the reference method, Y (dependent variables). This technique creates new variables to decrease data dimensionality called principal components (PCs) [72]. In order to verify whether the model was robust in predicting new samples, cross-validation was used. In this case, a sample is taken from the calibration set and the model is created with the rest of the model samples. Thus, the model parameters do not change significantly when new samples are added to the calibration set and can be applied to complex mixtures [69].

Statgraphics Centurium XV (Manugistics Inc., Rockville, MD, USA) was used for comparisons between LC-MS/MS and NIR. Normality of the residues was verified by the Shapiro-Wilk test. The Student's t (parametric variables) and the Mann-Whitney (non-parametric variables) tests were applied to compare the methods. The significance threshold was set at 0.05.

## Evaluated parameters

Model performance was assessed by correlation coefficient (R), determination coefficient ($R^2$) (the higher the $R^2$, the better the model), root mean square error of calibration (RMSEC) and validation (RMSEP), standard error of calibration (SEC) and standard error of prediction (SEP), which is based on the residuals. This is the difference between the predicted values and the actual values of the *n* samples of the calibration set. Residues represent the information contained in the *n* reference sample data that is not explained by the model. Complete validation of the model involves the study of the validation set. The samples in this set are used to test the predictive quality of the model by calculating R, $R^2$ and SEP; the last performance metric to be reported is the residual prediction deviation (RPD), which represents the model's ability to predict unknown samples, considering the variability of the set. In addition, the variables (wavelengths) that contributed to describe the most important differences between the samples, based on the PCs, were investigated.

## Results

### Samples and spectral information

This study analyzed 676 maize samples, being 236 for FBs and 440 for ZEN. The spectral range includes the visible region (400–100 nm) and the near infrared region (1100–2500 nm). The models were built separately for the each of the mycotoxins assessed. The sum of $FB_1$ levels varied from 125 to 24,200 $\mu g.kg^{-1}$; mean value and standard deviation (SD) were 5,643 $\mu g.kg^{-1}$ and 5,666, respectively. The sum of $FB_2$ levels varied from 125 to 9,210 $\mu g.kg^{-1}$; mean value and SD were 2,263 $\mu g.kg^{-1}$ and 2,279, respectively. ZEN database ranged from 20 to 884 $\mu g.kg^{-1}$; mean value and SD were 103 $\mu g.kg^{-1}$ and 151, respectively.

### Raw data processing

The regression method used in the model was PLS, using cross-validation in the three models developed ($FB_1$, $FB_2$ and ZEN). Several individual and combination mathematical treatments

have been investigated, such as smoothing, normalization (mean, maximum and range), baseline offset, multiplicative Scatter Correction (MSC), derivatives, standard normal variate (SNV) and detrending (DT); the chosen model was the one that provided the best accuracy. Fig 1 shows the higher positive loads and the wavelengths which explain the data variance, being represented by PC 1 and PC 2. These first two PCs carry the greatest information about the model. The higher the loading in a given wavelength, the more important this variable is. For example, for FBs, the wavelength range 1900–2498 nm is the most important for having the higher loading. For PC 2, the most important ranges are from 400 to 500 nm and some specifics bands (2100, 2200 and 2450 nm). Regarding ZEN, variables between 400–500 and 2100–2400 nm present the higher loads; for PC 2, ranges varying between 400–500 and 1200–1900 nm demonstrate to be the most important.

Several mathematical treatments were evaluated in order to remove irrelevant spectral information and enhance accuracy in FBs and ZEN calibrations. For FBs, the spectral pre-processing in which the best accuracy was obtained was the Savitski-Golay algorithm (9-point window, 2nd degree polynomial and 2nd derivative). The PLS analysis chosen was the model with 18 PCs for $FB_1$ and 10 PCs for $FB_2$. Fig 2 shows a representation of the explained calibration variance; 100% of the data variation is explained with 18 and 10 PCs in calibration and 73 and 71% in validation, respectively.

For ZEN, the best calibration model was obtained by using smoothing Savitski-Golay algorithm. The PLS analysis chosen was the model with 7 PCs. Fig 3 shows a representation of the

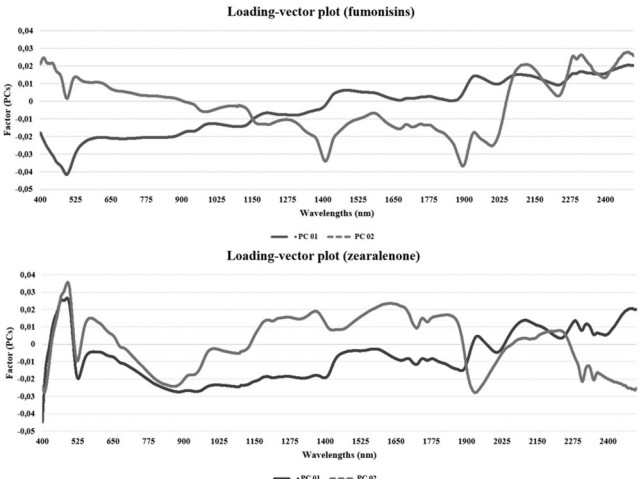

**Fig 1. Graph showing important variables.** Representation of the most important variables based on the first principal components (PCs), PC 1 and PC 2, for fumonisins and zearalenone in maize samples.

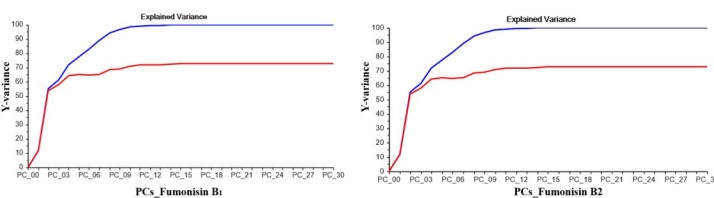

**Fig 2. Explained variance of fumonisin $B_1$ ($FB_1$) and fumonisin $B_2$ ($FB_2$):** Plot of explained calibration and validation variance versus number of Principal Components (PCs) for $FB_1$ and $FB_2$.

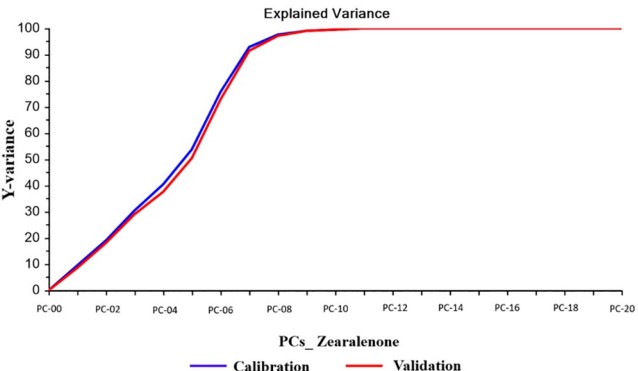

**Fig 3. Explained variance of zearalenone: Plot of explained calibration and validation variance versus number of Principal Components (PCs) for zearalenone.**

explained calibration variance; 93% of the data variation is explained with 7 PCs in calibration and 91% in validation.

R, $R^2$, RMSEC, SEC, RMSECV and RPD (the ratio of SD and SECV) were, respectively: for $FB_1$, 0.993, 0.987, 588, 586, 2,793 and 2.028; and for $FB_2$, 0.992, 0.984, 258, 258, 1,137 and 2.004. The ability of the calibration and validation model was assessed by comparing the reference results (LC-MS/MS) with the values predicted by NIR (S1 and S2 Figs).

For ZEN the parameters R, $R^2$, RMSEC, SEC, RMSECV and RPD (the ratio of SD and SECV) were, respectively: 0.926, 0.962, 41.07, 41.11, 44.64 and 3.382. The ability of the calibration and validation model was assessed by comparing the reference results (LC-MS/MS) with the values predicted by NIR (S3 Fig).

## Validation using unknown samples

In order to assess the accuracy of the model, 200 unknown samples were predicted (100 for FBs and 100 for ZEN). The selection process for unknown samples followed the same procedure as that developed in the calibration model. After reading the spectra, the samples were predicted. The external validation results were expressed as total FBs ($FB_1+FB_2$). FBs levels varied from 250 to 12,700 μg.kg$^{-1}$; mean value and SD were 2,690 μg.kg$^{-1}$ and 2,275, respectively. R, $R^2$, RMSEP, SEP and RPD were 0.809, 0.899, 659, 682 and 3.33, respectively. The prediction results were compared with the results of the reference levels (LC-MS/MS) and analyzed statistically using Student's t test (p = 0.32), indicating a good predictive ability. The results found in the prediction were reported in Fig 4.

ZEN levels varied from 20.0 to 834.8 μg.kg$^{-1}$; mean value and SD were 123.5 μg.kg$^{-1}$ and 189.1, respectively. R, $R^2$, RMSEP, SEP and RPD were 0.991, 0.984, 69.7, 69.8 and 2.71, respectively. The prediction results were compared with the results of the reference levels (LC-MS/MS) and statistically analyzed using Mann-Whitney test (p = 0.18), demonstrating that there is no statistical difference between the methodologies. Results found in the prediction were reported in Fig 5.

## Discussion

Interpretation of spectra related to fungal compounds and mycotoxins is rather complex, since they occur at low concentrations in the cereals. Moreover, the great variability of chemical compounds present may lead to band overlapping [63], which makes direct determination difficult. When attacked by fungi, maize grains lose important nutrients such as proteins, fats and

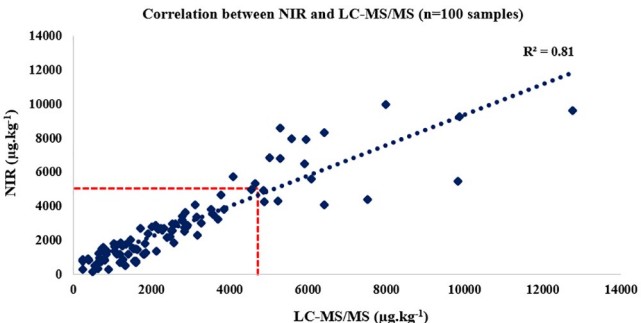

**Fig 4. External validation of fumonisins: Correlation between Near Infrared Spectroscopy (NIR) and Liquid Chromatography Coupled to Tandem Mass Spectrometry (LC-MS/MS) in ground maize.**

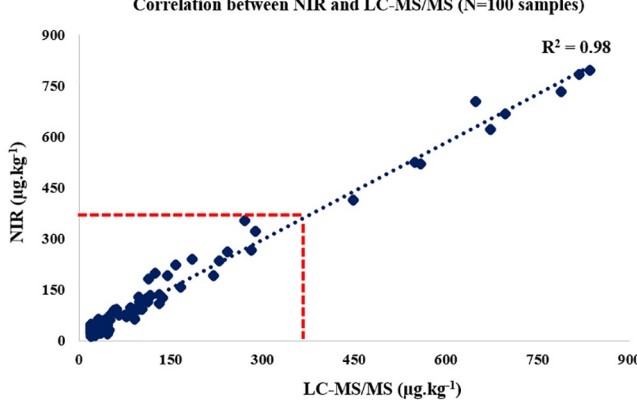

**Fig 5. External validation of zearalenone: Correlation between Near Infrared Spectroscopy (NIR) and Liquid Chromatography Coupled to Tandem Mass Spectrometry (LC-MS/MS) in ground maize.**

vitamins, thus causing spectral alterations. These changes can be detected by NIR through mathematical treatments that amplify the information. In this way, all wavelengths were used for the development of calibrations in the present study, since it is not clear which ones have the capacity of mycotoxicological diagnosis [63,70]. Nonetheless, investigation of the most important variables indicates that the most relevant ranges for FBs are within 400–600 and 1900–2500 nm. For ZEN, the wavelength ranges of 400–500, 1200–1900 and 2100–2400 nm are the most important. Similar findings have been noted when exploring the discriminant analysis for FBs in maize [58].

In the calibration database, 100 samples presented FBs levels above 5,000 μg.kg$^{-1}$, whereas the remaining 136 elements did not exceed this threshold. For ZEN, 27 samples were above the standards of the Brazilian legislation (400 μg.kg$^{-1}$). The database used to construct the predictive model was quite representative and adequate, since it covered a wide range of FBs and ZEN concentrations. It is essential to build a database inserting samples based on different regions, since the levels and frequency of contamination may vary depending on the climatic conditions of each region [73]. Furthermore, fungal development and consequent mycotoxin production may occur despite the implementation of good production practices [74]. Global levels of FBs in unprocessed cereals, including maize, range from 39% in Europe to 95% in America [75]. For ZEN, the incidence reported in contaminated food-crops worldwide has a

large variation; the average contamination is 30–40%, ranging from 15% in Asia to 59% in Africa [73, 75].

During calibration development, some samples were automatically excluded as they were considered outliers by the development software [76]. Exclusion of such data is important because their inclusion can negatively affect the model and prediction errors may occur. At the end of this process, the model was reduced to 203 samples for $FB_1$ and 202 samples for $FB_2$, *i.e.*, 33 and 32 samples were excluded, respectively. When building ZEN model, no sample was excluded from the calibration.

As shown in S1 and S2 Figs, the model allowed separation of almost all samples with FBs content < 5,000 μg.kg$^{-1}$ from those containing > 5,000 μg.kg$^{-1}$, although seven samples were incorrectly estimated and thus considered false negatives. No sample was considered a false positive. In the calibration results for ZEN, considering the legal limit of 400 μg.kg$^{-1}$, two samples were considered false negatives and one was a false positive.

A number of PCs of FBs similar to that found in this study was selected as the best calibration model in an investigation on FBs in ground maize using Fourier Transform Near Infrared Spectroscopy (FT-NIR) (PCs = 18) [70]. In another assessment carried out in maize samples, the authors found the best model using 17 PCs [77]; the chosen model had an acceptable accuracy in relation to the content of FBs as well as a good predictive capacity in the evaluation of unknown samples.

Selecting the ideal number of PCs is paramount for the quality of a model [78]; using a smaller number may provide unsatisfactory results as not all available data is used. On the other hand, if a large number of latent variables is included, it may evidence a deterioration of the analysis by incorporating overfitting [79]. In the current study, a higher PC value is due to the data structure, *i.e.*, because there is a large number of variables, the matrix becomes complex and cannot be explained by a small number of components [76].

The RPD of 3.33 found in the external validation indicated a good predictive ability. Calibration models with RPD>2 and $R^2$>0.80 are considered satisfactory [80]. Additionally, focusing on values around 5,000 μg.kg$^{-1}$, five elements were misclassified according to the legal level, 2% being classified as false positives and 3% as false negatives. In an assessment conducted with FBs on maize, the authors observed three incorrectly predicted samples, considering the 4,000 μg.kg$^{-1}$ guidance level of the European legislation [70]. So, the developed model ensures a good screening ability.

Another work examined the use of NIR in the analysis of fungal infection (*F. verticillioides*), ergosterol and $FB_1$ in maize samples [63]; it was concluded that NIR can be applied for monitoring post-harvest fungal contamination as well as for distinguishing contaminated lots.

ZEN RPD and $R^2$ values of 2.71 and 0.98, respectively, indicate that satisfactory results of prediction of ZEN with unknown samples. Furthermore, no false positive or false negative results were observed. The use of NIR to predict ZEN has been investigated in other ingredients. In Southern Brazil, wheat kernel and milled wheat samples naturally infected by *Fusarium graminearum* were analyzed by infrared spectroscopy, and the $R^2$ values of 0.86 and 0.87 as well as the SECV levels of 254.29 and 231.85 μg.kg$^{-1}$ found for such products, respectively, using Multivariate Partial Least Squares (MPLS) regression represented an acceptable prediction of ZEN content by NIR [81].

## Conclusions

This is the first study providing scientific knowledge on the determination of FBs and ZEN in Brazilian maize samples using the NIR technology. Maize is a global commodity, being routinely used to produce feed and food. This paper reveals the potential of NIR as a fast and easy

methodology for predicting FBs and ZEN in this truly important cereal compared to conventional techniques. Good correlation between measured and predicted values proved the reliability and accuracy of the model for future samples. Nevertheless, as there may be variations in FBs and ZEN levels between different climates and regions, calibrations must be constantly updated. While traditional methods are generally expensive, complex and require several working days to be finalized, NIR is an inexpensive, environmentally friendly, and fast procedure that only needs a few minutes for spectra collection and FBs and ZEN content prediction. This tool also allows the analysis of a larger number of samples and thus the prior control of cereal batches with contamination above the limit of the local legislation.

## Supporting information

**S1 Fig. Fumonisin B$_1$ (FB$_1$): Predictive partial least squares model with 18 Principal Components (PCs)–linear regression plot of measured and estimated concentrations (µg.kg$^{-1}$).**
(TIF)

**S2 Fig. Fumonisin B$_2$ (FB$_2$): Predictive partial least squares model with ten Principal Components (PCs)–linear regression plot of measured and estimated concentrations (µg.kg$^{-1}$).**
(TIF)

**S3 Fig. Zearalenone (ZEN): Predictive partial least squares model with seven Principal Components (PCs)–linear regression plot of measured and estimated concentrations (µg.kg$^{-1}$).**
(TIF)

## Acknowledgments

The authors would also like to thank the Federal University of Santa Maria (UFSM), Coordination for the Improvement of Higher Education Personnel (CAPES) and LAMIC for enabling the development of this project.

## Author Contributions

**Investigation:** Denize Tyska, Adriano Olnei Mallmann, Juliano Kobs Vidal, Carlos Alberto Araújo de Almeida, Carlos Augusto Mallmann.

**Methodology:** Denize Tyska, Adriano Olnei Mallmann, Juliano Kobs Vidal, Carlos Alberto Araújo de Almeida, Luciane Tourem Gressler, Carlos Augusto Mallmann.

**Resources:** Luciane Tourem Gressler.

**Supervision:** Luciane Tourem Gressler.

**Writing – original draft:** Denize Tyska, Adriano Olnei Mallmann, Juliano Kobs Vidal, Carlos Alberto Araújo de Almeida, Luciane Tourem Gressler, Carlos Augusto Mallmann.

**Writing – review & editing:** Denize Tyska, Adriano Olnei Mallmann, Juliano Kobs Vidal, Carlos Alberto Araújo de Almeida, Carlos Augusto Mallmann.

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
