## [Decision Letter · Decision Letter 0]

22 Apr 2020

PONE-D-20-04995

Multivariate method for prediction of fumonisins B1 and B2 in maize using Near Infrared Spectroscopy (NIR)

PLOS ONE

Dear authors

Thank you for submitting your manuscript to PLOS ONE. After careful consideration, we feel that it has merit but does not fully meet PLOS ONE’s publication criteria as it currently stands. Therefore, we invite you to submit a revised version of the manuscript that addresses the points raised during the review process.

A major concern from one of the reviewers is the novelty of the work and I have to agree with the reviewer. Nevertheless, we want to give you the opportunity to clarify more specifically what is new in your data compared to data published by other research groups in the past.

In addition, also take into account the comments raised by reviewer 1 on the clarity of the discussion

We would appreciate receiving your revised manuscript by 21/05/2020. To enhance the reproducibility of your results, we recommend that if applicable you deposit your laboratory protocols in protocols.io, where a protocol can be assigned its own identifier (DOI) such that it can be cited independently in the future. For instructions see: http://journals.plos.org/plosone/s/submission-guidelines#loc-laboratory-protocols

We look forward to receiving your revised manuscript.

Kind regards,

Kris Audenaert, Ph.D.

Academic Editor

PLOS ONE

Journal Requirements:

Reviewers' comments:

Reviewer's Responses to Questions

**Comments to the Author**

1. Is the manuscript technically sound, and do the data support the conclusions?

Reviewer #1: Yes

Reviewer #2: Yes

2. Has the statistical analysis been performed appropriately and rigorously? 

Reviewer #1: Yes

Reviewer #2: Yes

3. Have the authors made all data underlying the findings in their manuscript fully available?

Reviewer #1: Yes

Reviewer #2: Yes

4. Is the manuscript presented in an intelligible fashion and written in standard English?

Reviewer #1: No

Reviewer #2: Yes

5. Review Comments to the Author

Reviewer #1: The paper describes the application of the Near Infrared Spectroscopy (NIR) to quantify the level of contamination of total fumonisins in ground maize, and its comparison with well assessed methods such as Liquid Chromatography Coupled Tandem Mass Spectrometry (LC-MS/MS), to verify the suitability of NIR to rapidly analyze a large numbers of maize samples for FBs contamination.

The paper describes experiments well carried out and the conclusions are correct, showing the reliability of this new methods.

However, the paper needs to be deeply revised for the English, that often is very difficult to understand. Also, the way to present data needs to be revised: . I would separate completely Results and Discussion sections in order to have more clear both parts, that in this version are extremely hard to follow.

Finally, minor revisions are reported below.

ABSTRACT

Line 27: the mycotoxins are not phytopathogens. The fungi that produce them are phytopathogens, not the metabolites produced. Eventually, they can be defined phytotoxic compunds.

Line28: Fusarium moniliforme is not acceptable at all as definition. It is an old nomenclature. Along the whole paper F. moniliforme must be replaced with F. verticillioides

INTRODUCTION

Line 47: Alternaria don’t produce fumonisins B

Line48: Fusarium moniliforme is not acceptable at all as definition. It is an old nomenclature. Along the whole paper F. moniliforme must be replaced with F. verticillioides

Line 50: the mycotoxins are not phytopathogens. The fungi that produce them are phytopathogens, not the metabolites produced. Eventually, they can be defined phytotoxic compunds.

Line 84: replace the word corn with the word maize and replace it along the paper

REFERENCE

Are the references below reported correctly?

Line 283: 1. RHEEDER J. Fusarium moniliforme and Fumonisins in Corn in Relation to Human Esophageal Cancer in Transkei. 1992;

Line 285 2. BACON CW, NELSON PE. Fumonisin Production in Corn by Toxigenic Strains of 286 Fusarium moniliforme and Fusarium proliferatum. J Food Prot [Internet]. 1994;57(6):514–21. Available from: https://doi.org/10.4315/0362-028X-57.6.514

Line 288 3. JACKSON, L.; JABLONSKI J. Fumonisins. In: MAGAN, N.; OLSEN M (Org. ., editor. Mycotoxins in food. Cambridge: Woodhead Publishing; 2004. p. 367–91.

Reviewer #2: The manuscript is written very clearly and appropriate methods are used to come to the results. However, it has one major drawback: The manuscript does not present any new information. If I compare the manuscript to that of e.g. Gaspardo et al. (2012, https://doi.org/10.1016/j.foodchem.2012.06.078). A similar method is used and I see exactly the same graphs and the descriptions are even very similar E.g. In Gaspardo et al. (2012) it was written: The data set was organised in a matrix with 143 rows and 926 + 1 columns, combining spectral and chemical data for each sample. In the current manuscript, it is written: The calibration set was arranged in a matrix with 236 rows and 4200 + 1 columns, combining spectral and chemical data for each sample. I can understand that this is indeed the way to deal with this information. But, the main issue is that I cannot really find new findings in this manuscript. You use the same methods for the same toxins in the same matrix. In addition, in 2018, Levasseur-Garcia, presented a clear review on this topic summarizing the toxins and matrices that could be measured with this method (doi: 10.3390/toxins10010038). In addition, the model presented in PONE-D-20-04995 seems to fit the data quiet well, but it is never mentioned which wavelengths or regions in the spectrum are important for toxin quantification.

I suggested a major revision, since I would like to give the authors the chance to significantly improve their work. Now it is merely a copy of what has been done already a lot of times. If they can include e.g. new toxins in a new matrix, or detect mixtures of toxins… or if they can compare several other machine learning technique or new deep learning methods, so that the manuscript had some added value compared to what has been published in the past, I am willing to review this manuscript.

6. PLOS authors have the option to publish the peer review history of their article (what does this mean?). If published, this will include your full peer review and any attached files.

Reviewer #1: No

Reviewer #2: No

---

## [Author Response · Author response to Decision Letter 0]

20 May 2020

Reviewer #1: 

The paper describes the application of the Near Infrared Spectroscopy (NIR) to quantify the level of contamination of total fumonisins in ground maize, and its comparison with well assessed methods such as Liquid Chromatography Coupled Tandem Mass Spectrometry (LC-MS/MS), to verify the suitability of NIR to rapidly analyze a large numbers of maize samples for FBs contamination.

The paper describes experiments well carried out and the conclusions are correct, showing the reliability of this new methods.

However, the paper needs to be deeply revised for the English, that often is very difficult to understand. Also, the way to present data needs to be revised: . I would separate completely Results and Discussion sections in order to have more clear both parts, that in this version are extremely hard to follow.

Finally, minor revisions are reported below.

The manuscript was thoroughly revised following the reviewer's suggestions; furthermore, results and discussion are presented separately.

ABSTRACT

Line 27: the mycotoxins are not phytopathogens. The fungi that produce them are phytopathogens, not the metabolites produced. Eventually, they can be defined phytotoxic compunds. 

This information was removed from the manuscript.

Line28: Fusarium moniliforme is not acceptable at all as definition. It is an old nomenclature. Along the whole paper F. moniliforme must be replaced with F. verticillioides

This information was removed from the abstract due to the changes made in the manuscript. 

INTRODUCTION

Line 47: Alternaria don’t produce fumonisins B

This information was removed from the manuscript.

Line48: Fusarium moniliforme is not acceptable at all as definition. It is an old nomenclature. Along the whole paper F.moniliforme must be replaced with F. verticillioides

Fusarium moniliforme has been replaced by Fusarium verticillioides in line 55 and throughout the text.

Line 50: the mycotoxins are not phytopathogens. The fungi that produce them are phytopathogens, not the metabolites produced. Eventually, they can be defined phytotoxic compunds.

This information was removed from the manuscript.

Line 84: replace the word corn with the word maize and replace it along the paper.

The word corn was replaced by maize throughout the manuscript.

REFERENCE

Are the references below reported correctly?

Line 283: 1. RHEEDER J. Fusarium moniliforme and Fumonisins in Corn in Relation to Human Esophageal Cancer in Transkei. 1992;

This reference has been adjusted (line 417).

Line 285 2. BACON CW, NELSON PE. Fumonisin Production in Corn by Toxigenic Strains of 286 Fusarium moniliforme and Fusarium proliferatum. J Food Prot [Internet]. 1994;57(6):514–21. Available from: https://doi.org/10.4315/0362-028X- 57.6.514

This reference has been adjusted (line 420).

Line 288 3. JACKSON, L.; JABLONSKI J. Fumonisins. In: MAGAN, N.; OLSEN M (Org. ., editor. Mycotoxins in food. Cambridge: Woodhead Publishing; 2004. p. 367–91.

This reference has been adjusted (line 467).

Reviewer #2: 

The manuscript is written very clearly and appropriate methods are used to come to the results. However, it has one major drawback: The manuscript does not present any new information. If I compare the manuscript to that of e.g. Gaspardo et al. (2012, https://doi.org/10.1016/j.foodchem.2012.06.078). A similar method is used and I see exactly the same graphs and the descriptions are even very similar E.g. In Gaspardo et al. (2012) it was written: The data set was organised in a matrix with 143 rows and 926 + 1 columns, combining spectral and chemical data for each sample. In the current manuscript, it is written: The calibration set was arranged in a matrix with 236 rows and 4200 + 1 columns, combining spectral and chemical data for each sample. I can understand that this is indeed the way to deal with this information. But, the main issue is that I cannot really find new findings in this manuscript. You use the same methods for the same toxins in the same matrix. In addition, in 2018, Levasseur-Garcia, presented a clear review on this topic summarizing the toxins and matrices that could be measured with this method (doi: 10.3390/toxins10010038). In addition, the model presented in PONE-D-20-04995 seems to fit the data quiet well, but it is never mentioned which wavelengths or regions in the spectrum are important for toxin quantification.

I suggested a major revision, since I would like to give the authors the chance to significantly improve their work. Now it is merely a copy of what has been done already a lot of times. If they can include e.g. new toxins in a new matrix, or detect mixtures of toxins… or if they can compare several other machine learning technique or new deep learning methods, so that the manuscript had some added value compared to what has been published in the past, I am willing to review this manuscript.

We are very grateful for the suggestions given to improve the manuscript. We read all the considerations and understood that the paper needed some changes.

Indeed, there are other studies using NIR to predict FBs in maize. However, they were carried out in different countries, with positivity and concentrations that may vary from one region to another. In Brazil, positivity of FBs in maize varies from 90 to 100%, and its inclusion in animal diet is quite high. Furthermore, in the poorest regions of the country, maize is the basis of daily meals, being widely consumed by the population. The use of quick tools, such as NIR, speeds up decision making by the Industry. So, we understand that this study is valid and important from an epidemiological point of view.

We followed your suggestion and included the assessment of another very important mycotoxin: zearalenone. This toxin has been little explored in maize using the NIR technology. Besides, it causes much damage to the reproductive system of farm animals, with significant impacts on production. There are also studies indicating changes in children, associating early puberty with maize-based food contaminated with zearalenone. As I mentioned above, there is a great human consumption of maize-based by-products.

It should also be highlighted that this is the first scientific examination carried out in Brazil using the NIR methodology to predict fumonisins and zearalenone in maize. We used samples from several Brazilian states to compose the database. Many studies are aimed at countries with different climates and frequencies of positivity.

Regarding the investigation of spectral bands, as mentioned in the manuscript, this information remains to be clarified, so we used the entire spectrum. However, work approaching this matter is under development by our research group and it will be promptly published.

Specific comments

L27: Fumonisins (FBs) are mycotoxins which are major phytopathogens => Fumonisins are no pathogens they are produced by phytopathogens.

This information was removed from the manuscript.

L28: produced mainly by Fusarium proliferatum and Fusarium moliniforme => F. verticillioides is also an important producer of Fumonisins

This information was removed from the abstract due to the changes made in the manuscript. 

L35: cross validation => cross-validation

Adjusted as suggested: lines 36, 193 and 242.

L38: No significant difference was observed between predicted values using NIR and reference values using Liquid Chromatography Coupled to Tandem Mass Spectrometry (LC-MS/MS) (p > 0.05). => This is only mentioned in the abstract, I never found it back in the results section, in addition which statistical test was used to test this?

All statistical tests used to make comparisons between LC-MS/MS and NIR are mentioned in lines 197 to 200. In the results section, the statistical test and the p value for total fumonisins are mentioned in lines 312-315; the statistical results of zearalenone and the p value are mentioned in lines 325 to 328.

L50: These 50 mycotoxins are chief phytopathogens of maize => Toxins are no pathogens

This information was removed from the manuscript.

Fig 5. What is the difference between the red and the blue line => I suggest calibration and validation, but it is not in the caption which color refers to calibration and which color to validation.

Fig 5 was adjusted and the captions were inserted: the blue line corresponds to calibration and the red line to validation.

L180: Several models were developed using different spectral treatments => Which models? Which spectral treatments?

Several calibration models have been developed using individual and combination mathematical treatments such as smoothing, normalization (mean, maximum and range), baseline offset, multiplicative Scatter Correction (MSC), derivatives, standard normal variate (SNV) and detrending (DT). The model with the best accuracy was selected as the finest one. This information was entered in the results section in lines 242-246.

L182 as they were considered outliers by the development software => What is defined as an outlier by the software? 

An anomalous sample, also known as an outlier, is an object or, in our case, a sample, which distances itself from the others within the sample set and may not belong to the same population as the majority. Generally, the presence of these samples allows the construction of models with high values of errors and low capacity of prediction, significantly influencing results prediction. Thus, the detection of outliers is of paramount importance and, often, the removal of these samples leads to the construction of more accurate and efficient models, ensuring their predictive quality. A sample can be considered as an outlier due to several reasons: measurement error, noise, extreme samples, deviating products or wrong tabling.

The software offers several regression methods for making models (calibrations). Regression is a generic term used for all methods that try to model and analyze several variables (in our case, 4,200 independent variables, X, and 1 dependent variable, Y) in order to build a relationship between two groups of variables. The adjusted model can then be used only to predict new values.

In regression, there are many ways for a sample to be classified as anomalous. It can be peripheral according to X variables only, or Y variables only, or both. Moreover, it may not be a discrepancy for a separate set of variables, but it can become a discrepancy when considering the relationship (X, Y). We used Partial Least Square (PLS) as a regression method. It is the most suitable for quantitative analysis and the calibrations are generally more robust and can be applied in complex mixtures, as in our case. The fundamental basis of this method is the Principal Component Analysis (PCA), which consists of manipulating the data matrix in order to represent the variations occurring in many variables, through a smaller number of "factors". From this, new systems of axes (called factors, main components, latent variables or even eigenvectors) are constructed to represent the samples, in which the multivariate nature of the data can be visualized in a few dimensions.

The software uses several methodologies to detect outliers, among which are the ones we used: "leverage" and Student residues. Leverages are useful for detecting samples which are far from the center within the space described by the model. Samples with high leverage differ from the average samples; in other words, they are likely outliers. A large leverage also indicates a high influence on the model. Student residues are the concentration residues, which are calculated in cross-validation, that is, poorly modeled samples have high residues.

L190 At the end of this process, the model was reduced to 203 samples for FB1 and 202 samples for FB2, i.e., 33 and 32 samples were excluded, respectively. => Which samples were excluded. Is there a reason why these samples were excluded? It is always interesting to have an in-depth look at the outliers.

Yes, it is important to look at these samples carefully, as in some cases it can be a measurement error. The software automatically excludes some samples when creating the model using specific detection tools (for example, leverage and student residue). In this study, the outliers were samples that presented very extreme results and were thus very distant from the population, or they were samples with different characteristics from the others in our database. In such cases, removing these samples is the best alternative. However, the calibrations are dynamic, so from time to time we insert new samples that are considered interesting to make the model more robust and expand the range of sample frequency so that all ranges of concentration and variability are represented.

---

## [Decision Letter · Decision Letter 1]

6 Nov 2020

PONE-D-20-04995R1

Multivariate method for prediction of fumonisins B1 and B2 and zearalenone in Brazilian maize using Near Infrared Spectroscopy (NIR)

PLOS ONE

Dear Dr. Mallmann,

Thank you for submitting your manuscript to PLOS ONE. After careful consideration, we feel that it has merit but does not fully meet PLOS ONE’s publication criteria as it currently stands. Therefore, we invite you to submit a revised version of the manuscript that addresses the points raised during the review process.

We look forward to receiving your revised manuscript.

Kind regards,

Vijai Gupta, PhD in Microbiology

Academic Editor

PLOS ONE

Additional Editor Comments (if provided):

MS needs additional corrections before it may be considered for publication in PLOS One. Kindly do the needful changes and submit your revised MS.

Reviewers' comments:

Reviewer's Responses to Questions

**Comments to the Author**

1. If the authors have adequately addressed your comments raised in a previous round of review and you feel that this manuscript is now acceptable for publication, you may indicate that here to bypass the “Comments to the Author” section, enter your conflict of interest statement in the “Confidential to Editor” section, and submit your "Accept" recommendation.

Reviewer #1: (No Response)

Reviewer #2: (No Response)

2. Is the manuscript technically sound, and do the data support the conclusions?

Reviewer #1: Yes

Reviewer #2: Partly

3. Has the statistical analysis been performed appropriately and rigorously? 

Reviewer #1: Yes

Reviewer #2: Yes

4. Have the authors made all data underlying the findings in their manuscript fully available?

Reviewer #1: Yes

Reviewer #2: No

5. Is the manuscript presented in an intelligible fashion and written in standard English?

Reviewer #1: Yes

Reviewer #2: Yes

6. Review Comments to the Author

Reviewer #1: I revised the first version of the paper and I think the authors much improved the original paper. Therefore I dont have any further criticism that would not allow the publication of the paper

Reviewer #2: ADVISE: MAJOR REVISION. Although the manuscript improved a lot compared to the previous version, there are still some important changes necessary before it can be published.

In your answer to my first question you mention that the novelty in the article lies in the fact that Many studies are aimed at countries with different climates and frequencies of positivity. Do you mean that samples with ZEN/FB from Brazil will be characterized by another spectrum of samples from another region (if the concentrations are similar)? In my opinion the shift in reflection caused by the presence of a certain toxin at a certain concentration is not highly depended on the region were you took the sample? Please explain in your discussion why it is important that you also do this analysis with samples from Brazil.

Another answer was: Regarding the investigation of spectral bands, as mentioned in the manuscript, this information remains to be clarified, so we used the entire spectrum. However, work approaching this matter is under development by our research group and it will be promptly published.

Ok you used the entire spectrum to fit the models, but based on the loadings of the PCA analysis and/or coefficients from the PLSR you can easily define characteristic bands for a certain toxin, I want to see this in the article before it can be published, otherwise it does not has an added value for other researchers, the only thing they now know is that you can predict these toxins. But it is important to compare these results with other findings….

In Figure 1 you show the spectrum of the fumonisin samples and of the zearalenone samples. It does not have an added value to divide it into “fumonisin” and “ZEN” samples as the samples you analysed for FB can also contain ZEN and vice versa (and also other toxins)?

Did you analyse certain samples for both ZEN and FB, since both toxins often co-occur?

In Figure 2 you show the results of the PCA analysis, however this figure is not very clear. Is it possible to make a 2D biplot and color the dots according to the concentration of ZEN/FB and then also draw the most important loadings so that the reader gains insight into which wavelengths are important for high ZEN or FUM concentrations

Can you include the models (coefficients of the PLS model) in supplementary data so that they can be tested by the readers?

I think you can reduce the number of figures, Figures 2-4 can be deleted, Fig. 5 replace it according to the previous comment.

Fig 8-10 can be in the supplementary data.

In the discussion you do not have to refer to figures

L365 ZEA => ZEN

7. PLOS authors have the option to publish the peer review history of their article (what does this mean?). If published, this will include your full peer review and any attached files.

Reviewer #1: No

Reviewer #2: No

---

## [Author Response · Author response to Decision Letter 1]

15 Dec 2020

PONE-D-20-04995R1

Authors’ responses to reviewers

Reviewer #1: 

I revised the first version of the paper and I think the authors much improved the original paper. Therefore I dont have any further criticism that would not allow the publication of the paper. 

• We greatly appreciate your time and effort regarding the revision of our manuscript.

Reviewer #2: 

1) In your answer to my first question you mention that the novelty in the article lies in the fact that Many studies are aimed at countries with different climates and frequencies of positivity. Do you mean that samples with ZEN/FB from Brazil will be characterized by another spectrum of samples from another region (if the concentrations are similar)? In my opinion the shift in reflection caused by the presence of a certain toxin at a certain concentration is not highly depended on the region were you took the sample? Please explain in your discussion why it is important that you also do this analysis with samples from Brazil. 

• It is important to develop calibrations of more specific databases for each region, since the contamination levels may vary from one place to another; so, samples originating from distinct regions and presenting diverse concentrations are required. In Brazil, for example, the levels of FBs are high, while the levels of ZEN are low. In Europe, in turn, the levels of ZEN and DON are high, while the levels of FBs are low. As suggested, this matter has been further addressed in the discussion section (lines 319 - 325).

2) Another answer was: Regarding the investigation of spectral bands, as mentioned in the manuscript, this information remains to be clarified, so we used the entire spectrum. However, work approaching this matter is under development by our research group and it will be promptly published. Ok you used the entire spectrum to fit the models, but based on the loadings of the PCA analysis and/or coefficients from the PLSR you can easily define characteristic bands for a certain toxin, I want to see this in the article before it can be published, otherwise it does not has an added value for other researchers, the only thing they now know is that you can predict these toxins. But it is important to compare these results with other findings…. 

• Indeed, we believe this information is relevant for future works, so we included our findings in the discussion section (lines 311-314).

3) In Figure 1 you show the spectrum of the fumonisin samples and of the zearalenone samples. It does not have an added value to divide it into “fumonisin” and “ZEN” samples as the samples you analysed for FB can also contain ZEN and vice versa (and also other toxins)? Did you analyse certain samples for both ZEN and FB, since both toxins often co-occur?

• The samples characterized by the spectra belong to different databases. The figure intended to represent the total amount of these samples and the ranges used. The samples included when building the models were tested in routine laboratory analysis, so not all of them were analyzed for FBs and ZEN simultaneously. Anyhow, this figure was removed from the manuscript.

4) In Figure 5 you show the results of the PCA analysis, however this figure is not very clear. Is it possible to make a 2D biplot and color the dots according to the concentration of ZEN/FB and then also draw the most important loadings so that the reader gains insight into which wavelengths are important for high ZEN or FUM concentrations.

• The image depicted in Figure 5 (now Figure 1) was replaced for a new one showing the most important variables, according to the reviewer’s suggestion.

5) Can you include the models (coefficients of the PLS model) in supplementary data so that they can be tested by the readers?

• This work has been developed along the past 10 years, and several development attempts have been tested. Thus, it is not possible to include the information regarding the coefficients of the PLS model, since it refers to confidential data.

6) I think you can reduce the number of figures, Figures 2-4 can be deleted, Fig. 5 replace it according to the previous comment.

• Figures 2-4 have been removed from the manuscript. Figure 5 was replaced by Figure 1, as suggested by the review.

7) Fig 8-10 can be in the supplementary data.

• Done as suggested (lines 382-389). 

8) In the discussion you do not have to refer to figures L365 ZEA => ZEN

• ZEA was replaced for ZEN in lines 76, 290 and 360.

---

## [Editor Report · Decision Letter 2]

21 Dec 2020

Multivariate method for prediction of fumonisins B1 and B2 and zearalenone in Brazilian maize using Near Infrared Spectroscopy (NIR)

PONE-D-20-04995R2

Dear Dr. Mallmann,

We’re pleased to inform you that your manuscript has been judged scientifically suitable for publication and will be formally accepted for publication once it meets all outstanding technical requirements.

Kind regards,

Vijai Gupta, PhD in Microbiology

Academic Editor

PLOS ONE

Additional Editor Comments (optional):

All the editorial, as well as reviewers comments, have been addressed.
---

## [Editor Report · Acceptance letter]

26 Dec 2020

PONE-D-20-04995R2 

Multivariate method for prediction of fumonisins B_1_ and B_2_ and zearalenone in Brazilian maize using Near Infrared Spectroscopy (NIR) 

Dear Dr. Mallmann:

I'm pleased to inform you that your manuscript has been deemed suitable for publication in PLOS ONE. Congratulations! Your manuscript is now with our production department. 

Kind regards, 

on behalf of

Dr. Vijai Gupta 

Academic Editor

PLOS ONE